# Natural Fiber-Reinforced Foamed Rubber Composites: A Sustainable Approach to Achieving Lightweight and Structural Stability in Sole Materials

**DOI:** 10.3390/polym17152043

**Published:** 2025-07-26

**Authors:** Yi Jin, Shen Chen, Jinlan Xie, Weixing Xu, Yunhang Zeng, Bi Shi

**Affiliations:** 1College of Biomass Science and Engineering, Sichuan University, Chengdu 610065, China; 15025036684@163.com (Y.J.); 17208294528@139.com (S.C.); 2National Engineering Laboratory for Clean Technology of Leather Manufacture, Sichuan University, Chengdu 610065, China; m17886985951@163.com (J.X.); zengyunhang@scu.edu.cn (Y.Z.); shibi@scu.edu.cn (B.S.); 3Research Center for Biomass Materials, Tianfu Yongxing Laboratory, Chengdu 610213, China; 4Key Laboratory of Leather Chemistry and Engineering, Sichuan University, Ministry of Education, Chengdu 610065, China

**Keywords:** sole material, natural rubber, natural fiber composite, foaming material, structural stability

## Abstract

Lightweightness and durability are key consumer demands for footwear. To address the issues of deformation and poor durability in foamed sole materials, this study integrates natural fibers into the formulation of foamed rubber. The effects of natural fiber incorporation on density, mechanical properties, creep behavior, anti-slip performance, and aging resistance were comprehensively analyzed. Additionally, the study explored the mechanisms underlying the improved performance of the modified rubber materials. The results revealed that natural fiber integration significantly enhanced the structural stability, strength, and aging resistance of natural rubber (NR). Among the fibers compared, collagen fibers (CF) proved to be the most effective modifier for foamed NR. The density, tensile strength, tear strength, and coefficient of friction of CF-modified foamed NR (CF-NR) were found to be 0.72 g/cm^3^, 10.1 MPa, 48.0 N/mm, and 1.105, respectively, meeting the standard requirements for sole materials. Furthermore, CF-NR demonstrated a recoverable deformation of 4.58% and a negligible irreversible deformation of 0.10%, indicating a successful balance between comfort and durability. This performance enhancement can be attributed to the supportive role of CF in the pore structure, along with its inherent flexibility and recoverability. This work presents a novel approach for the development of high-quality, lightweight footwear in the sole material industry.

## 1. Introduction

As living standards rise, footwear expectations increase. In recent years, consumers have increasingly prioritized lightness and comfort in their footwear choices [1]. Currently, polyurethane (PU) and ethylene-vinyl acetate (EVA) copolymer are the leading lightweight sole materials on the market [2,3]. While these materials excel at reducing weight, they fall short in low-temperature resistance, anti-slip properties, and aging performance [4]. In contrast, rubber is renowned for its exceptional elasticity, shape retention, low-temperature resilience, and anti-slip properties, solidifying its indispensable role in sole materials [5,6,7,8,9]. However, the higher density of rubber compared to EVA and PU limits its applicability in lightweight sole materials. To address this issue, various innovative strategies have been explored to reduce the weight of rubber. One of the most effective approaches is foaming, which reduces rubber’s density while conserving material and enhancing properties such as thermal insulation and impact load absorption [10]. This technique preserves the inherent qualities of rubber, achieving the dual goals of lightweight and comfortable footwear. Rubber foaming methods are typically categorized into physical and chemical foaming [11,12]. Physical foaming refers to the process of injecting inert gases, low-boiling-point liquids, and other substances into the polymer melt under high pressure and then expanding them into bubbles through heating or depressurization [13,14]. However, it has the disadvantages of a complex process, cumbersome operation, high cost, and poor foaming effect, so its application is limited [13,14]. Chemical foaming, favored for its simplicity and ease of implementation, is the dominant approach in practice. In natural rubber (NR), azodicarbonamide (AC) is commonly employed as the chemical foaming agent. AC is preferred for its high gas output, consistent product quality, non-corrosiveness, and straightforward production process, making it widely applicable [15]. However, this method can significantly reduce the tensile and tear strength of rubber [16], and it often exacerbates the material’s creep behavior, which compromises the durability of the sole material.

Amid the global push for carbon neutrality, research into natural fiber composites has garnered significant attention [17,18]. Natural fibers are environmentally friendly materials that offer excellent biodegradability and renewability [19]. In comparison to synthetic fibers, natural fibers present several key advantages, including affordability, superior mechanical properties, and effective thermal and acoustic insulation [20]. Moreover, when used as fillers in polymers, natural fibers generally maintain the highly ordered structure inherent in natural products, which can enhance the mechanical properties of the base material [21,22,23,24,25]. In the context of foamed materials, these fibers serve as heterogeneous nucleation sites for chemical foaming, improving both the foam ratio and uniformity of the foaming process [11]. Integrating natural fibers into a rubber foaming system promises not only to create lighter and more comfortable sole materials but also to enhance their overall performance [11,26]. In this study, bamboo fiber (BF)—noted for its inherently low-carbon footprint—together with collagen fiber (CF) and distiller’s spent-grain rice-husk fiber (DF) recovered from light-industry solid waste were selected as the raw materials [27,28,29,30]. After pulverization and silane modification, these fibers were incorporated into NR and its additives, resulting in foamed and molded composites. The study examined how the introduction of natural fibers affected the foaming behavior, mechanical properties, and structural stability of NR-based composites. The findings demonstrated that CF successfully balanced comfort and durability in NR-based sole materials. This work offers valuable insights into the design of high-quality, lightweight foamed materials for sole applications.

## 2. Materials and Methods

### 2.1. Materials

A total of 3 L grade of natural rubber (NR) was obtained from Zhongxiang International Group Co., Ltd. The foaming agents azodicarbonamide (AC), tetramethyl thiuram disulfide (TMTD), sublimed sulfur (S), and stearic acid were all AR grade and purchased from Sinopharm Chemical Reagent Co., Ltd. (Shanghai, China). Zinc oxide (ZnO, AR grade), sodium carbonate (Na_2_CO_3_, AR grade), and the silane coupling agent KH-550 (AR grade) were all AR grade and acquired from Chengdu Chemical Reagent Cologne Co. Ltd. (Chengdu, China). Bamboo fibers (BF) were supplied by Sichuan Changsheng New Material Technology Co. Ltd. (Yibin, China) Leather shavings were obtained from Xingye Leather Technology Co. Ltd. (Jinjiang, China), and distiller’s dried grains with solubles (DDGS) were procured from Luzhou Laojiao Co. Ltd. (Luzhou, China).

### 2.2. Preparation of Modified Natural Fibers

Leather shavings were initially immersed in deionized water at a 1:4 ratio (leather shavings to water), with the pH adjusted to 6.5–7.0 using a 0.1 mol/L sulfuric acid solution.

Distillers dried grains with solubles (DDGS) were immersed in a boiling water bath at a 1:10 ratio (DDGS to water) for 4 h, followed by filtration through an 18-mesh sieve to remove impurities such as pit mud, residual protein, and starch.

Subsequent to the drying process, commercial bamboo fibers, pre-processed leather shavings, and DDGS were subjected to grinding in a super centrifugal grinder (Retsch, ZM200, GER) until the particle size was smaller than 60 mesh, thus obtaining BFs, CFs, and DFs.

KH550, in a quantity equivalent to 10 wt% of the natural fibers, was initially dissolved in an ethanol-water mixture with a ratio of 93:7 at a KH550-to-mixture ratio of 1:10. The solution was stirred for 30 min at room temperature. The natural fibers were then immersed in this prepared solution and stirred for 12 h at room temperature. After filtration to remove the excess solution, the fibers were dried at 80 °C, yielding the modified natural fibers. The modified natural fibers exhibited an increase in N content or Si content tested by the energy-dispersive X-ray spectroscopy (EDS; Xplore 30, Oxford, UK), confirming the successful grafting of KH550 (Appendix A).

### 2.3. Preparation of Pure NR and Composites

Surface-modified natural fibers and NR were compounded in a torque rheometer (RM-200C, Harper, Harbin, China) at 60 rpm, with formulations detailed in Table 1. The process was initiated by masticating NR at 90 °C for 3 min. Subsequently, fiber reinforcements (BF, CF, DF) were incorporated and kneaded for 2 min. Stearic acid, ZnO, and foaming agent AC were sequentially introduced at 120 s intervals. The compound was then transferred to an open two-roll mill for 20 cycles of blending with TMTD and sulfur at 90 °C. The mixture was then vulcanized using a plate vulcanizer (PBLH-25, Jiezhun, Shanghai, China) under 5 MPa. The vulcanization process was set to 150 °C for 3 min based on the results obtained from the rubber vulcanization curves (Appendix A). The mechanical properties of the composites were determined under different fiber conditions, thereby establishing the reasonable fiber addition (Appendix A). The final products included pure NR (Blank) and composites (AC-NR, BF-NR, CF-NR, and DF-NR).

### 2.4. Characterization

The basic physical and mechanical properties were tested according to ISO 2781:2018 [31], ISO 37:2017 [32], and ISO 34-1:2022 [33], respectively. For the stress relaxation testing of the material, a compression elastic recovery tester (ZL-3026, Zhongli, China) was employed. Under room temperature (25 °C), the deformation associated with a stress level of 2.0 MPa was held constant, and the variation in stress was meticulously monitored and recorded.

To rapidly reveal the differences in molecular chain relaxation among different samples, creep-recovery curves and cyclic strain-relaxation curves obtained from DMA were tested at 50 °C according to the time-temperature superposition principle [25]. Creep-recovery properties of pure NR and composites were characterized via dynamic mechanical analysis (DMA Q850, TA Instruments, Newcastle, Delaware, USA) employing a three-point bending configuration at 50 °C. Strain evolution was acquired under fixed-frequency (1 Hz) and constant static stress (0.5 MPa) conditions. The test protocol maintained 1800 s loading/unloading intervals with creep strain monitored throughout the experimental duration. Numerical simulations were conducted using the Burger’s model for creep and the Weibull distribution function for recovery. Detailed information on the numerical simulations is shown in Appendix A

The cyclic stress-strain and stress relaxation curves of pure NR and composites were tested using DMA in strain mode at 50 °C. Pure NR and composites s were loaded at a rate of 5%/min until the deformation reached 20%, and then the deformation was recovered at a rate of 5%/min until the strain was reduced to zero. The above procedures were repeated 10 times to obtain 10-cycle stress-strain curves. The dynamic mechanical properties of the pure NR and composites were characterized using DMA in three-point bending mode. Testing spanned from −80 °C to 80 °C with a heating rate of 5 °C/min, under a constant frequency of 1 Hz and an oscillation amplitude of 1 mm.

The Fourier Transform Infrared (FT-IR) spectra of pure NR and composites were acquired by employing an FT-IR Spectrometer (NICOLET iS10, Thermo Fisher Scientific, Waltham, Massachusetts, USA). The C 1s, O 1s, and S 2p spectra of pure NR and composites were captured using an X-ray photoelectron spectroscope (XPS; Thermo 250Xi, Thermo Fisher Scientific, Waltham, Massachusetts, USA) (the detailed parameters of the measurement are located in Appendix A). With samples weighing 5–10 mg, the thermal decomposition behaviors of pure NR and composites were analyzed using a thermal gravimetric analyzer (TGA; TG 209 F1, Netzsch, Selb, GER) from 50 °C to 800 °C at a heating rate of 10 °C/min under N_2_ atmosphere. Following gold sputtering, the morphologies of different fibers and composites were observed using a Field Emission Scanning Electron Microscope (FESEM; Nova Nano-SEM450, FEI, Portland, Oregon USA) with a voltage of 5 kV. Pore-size distributions of pure NR and composites were determined with a mercury intrusion porosimeter (MIP; AutoPore V 9600, Micromeritics, Norcross, Georgia, USA). The crosslinking degree and crosslinking density of pure NR and composites were evaluated in accordance with ASTM D6814-18. The specific test parameters and calculation formulas are provided in Appendix A.

The friction coefficients of pure NR and composites were measured using a friction coefficient tester (MXD-01, Labthink, Jinan, China) in accordance with ISO 8295:1995 [34], with a slider mass of 200 g and a pulling speed of 100 mm/min. The tensile strength, tear strength, and elongation at break of pure NR and composites, following 100 h of treatment at 100 °C, were employed to assess the thermal-oxidative aging behaviors of the materials.

## 3. Results and Discussion

### 3.1. Effect of Natural Fiber Addition on the Mechanical Properties of Composites

Figure 1a and Appendix A present the apparent density and foaming ratio of pure NR and composites. The addition of AC substantially lessened the density of NR. Upon the addition of 20 wt% natural fibers, although there was an increase in material density, the densities of the composites still decreased by 25.21%, 24.33%, and 33.52%, respectively, compared to Blank. These findings indicate that the incorporation of natural fibers had a minimal effect on the lightweight modification of NR.

The mechanical properties of pure NR and composites are shown in Figure 1b,c. A noticeable decline in mechanical properties occurred after foaming, attributed to the reduction in material uniformity and the increased stress concentrations from foam pores. However, the addition of fibers resulted in an increase in tensile strength, tear strength, and elongation at break. This improvement can be attributed to the natural fibers enhancing force and energy transmission while restricting the mobility of NR molecular chains and minimizing irreversible intermolecular slippage.

### 3.2. Effect of Fibers on the Structural Stability of Composites

The viscoelastic properties of polymers make them prone to creep deformation, particularly under prolonged loading and elevated temperatures [25]. In foamed rubber, the stress concentrations caused by foam pores increase its susceptibility to creep, significantly compromising its structural stability and shortening its service life.

The failure of sole materials is typically attributed to creep failure (or creep rupture) under prolonged loading [35,36,37]. This phenomenon arises from the irreversible sliding of polymer chains, breaking of cross-linking points, or viscoelastic relaxation within the rubber under sustained stress. These mechanisms lead to a gradual loss of elasticity and the accumulation of permanent deformation in the material. The exacerbation of short-term creep (initial creep) can quickly lead to premature creep failure (accelerated creep). Given the substantial differences observed in the initial creep behavior across samples, further investigation into accelerated creep was deemed unnecessary.

According to material creep theory, a lower value of ${Ɛ}_{\infty }^{*}$ indicates a more stable material structure, with reversible deformations more likely. Conversely, a higher proportion of $\theta_{SK}^{*}$ in the total deformation correlates with enhanced material rebound capability. Since the progressive thinning of sole materials over time is primarily due to the accumulation of irreversible deformations, maintaining a significant level of elastic deformation is crucial for preserving comfort. Therefore, an ideal sole material should possess a sufficiently low ${Ɛ}_{\infty }^{*}$ and an appropriate $\theta_{SK}^{*}$.

The creep-recovery curves and deformation values of pure NR and composites (Figure 2a and Table 2) demonstrate that the addition of AC significantly increased all types of deformation. This occurs because the deformation of foam pores replaces the orientation and sliding of NR molecular chains under testing conditions. Additionally, stress concentrations due to foaming make NR molecular chains more prone to irreversible slippage. As a result, AC-NR is not suitable as an ideal sole material.

The incorporation of natural fibers effectively altered the creep behavior of the material. Notably, the ${Ɛ}_{\infty }^{*}$ of these composites decreased by more than 50% compared to that of Blank, which was considerably lower than that of AC-NR. Among the three natural fiber-modified composites, CF-NR exhibited the lowest ${Ɛ}_{\infty }^{*}$ and the highest $\theta_{SK}^{*}$, demonstrating superior resilience and structural stability. These attributes make CF-NR particularly well-suited for the development of lightweight sole materials.

To further investigate the effects of foaming and the incorporation of natural fibers on the movement behavior of rubber molecules, the creep and recovery curves of pure NR and composites were fitted using the Burger’s model and Weibull distribution. The results are presented in Appendix A, with the specific characteristic parameters listed in Table 3 and Table 4. (All the specific characteristic parameters are defined in Appendix A.) According to the Burger’s model, $\eta_{M}$ is associated with non-recoverable deformation. The $\eta_{M}$ values for the composites were all higher than that of AC-NR, indicating a reduction in non-recoverable deformation in the composites. $E_{M}$ represents the elastic deformation resulting from changes in the molecular internal chain length and chain angle. This deformation is fully recoverable once the external force is removed and is related to the elastic modulus of the molecular chain. Following fiber incorporation, $E_{M}$ increased significantly, suggesting an enhancement in the elastic modulus of the composites. $E_{K}$, $\eta_{M}$, and *τ* are linked to the rigidity of the amorphous molecular chain segments of NR in the short term. These parameters decreased notably after foaming, indicating that foaming diminished the creep resistance of NR. The introduction of BF and DF led to a considerable increase in these properties, demonstrating that BF and DF significantly improved the creep resistance of NR. While CF did not show notable modifications during the creep process, simulation results based on the Weibull distribution equation revealed its exceptional advantages during the recovery phase. With a near-zero ${Ɛ}_{\infty }$ (0.01%) and an ${Ɛ}_{R}$ (4.57%) comparable to that of Blank, CF-NR emerged as an ideal raw material for lightweight sole materials. The enhanced performance can likely be attributed to CFs superior recoverability and flexibility when compared to BF and DF.

Stress relaxation refers to the gradual decay of stress within a material under constant deformation and temperature over time and is considered a form of generalized creep [38,39]. To assess the effectiveness of natural fiber modification in real applications, the stress relaxation behavior of all samples was evaluated at room temperature (25 °C) using a compression resilience tester. The results are shown in Figure 2b. Under a deformation corresponding to a stress level of 2.0 MPa, all samples exhibited notable stress relaxation. The stress of Blank and AC-NR decreased by 43% and 48%, respectively, within 240 min after loading. In contrast, the incorporation of natural fibers significantly improved stress retention, with samples maintaining more than 67% of their initial stress after 300 min of loading. Notably, CF-NR demonstrated outstanding performance, with stress reduction of only 11.7%. These results underscore that natural fibers can significantly enhance the structural stability of sole materials in practical applications.

For sole materials, the stress experienced is not constant but cyclical. The soles undergo cyclic compression and recovery with each step. This behavior was simulated through cyclic tension-recovery tests using DMA, as shown in Figure 3. Blank and AC-NR exhibited negative stress when the deformation returned to 0. This behavior resulted from the instability of the structures in these two materials, leading to irreversible deformation during the first cycle and causing the material to experience reverse stress during recovery. As the cycles progressed, the molecular chains and fiber fillers in the material aligned in the direction of the applied stress, resisting tensile deformation. This resulted in an increase in initial stress as the number of cycles increased. The peak stresses for Blank, AC-NR, BF-NR, CF-NR, and DF-NR increased by 0.104 MPa, 0.128 MPa, 0.249 MPa, 0.609 MPa, and 0.128 MPa, respectively. Among them, the CF-NR specimen exhibited the greatest increase in stress (88.57%) and demonstrated the best resistance to deformation. Therefore, sole materials incorporating CF-NR are likely to show improved resilience and longer service life in practical applications, making them a more durable option.

### 3.3. Mechanism of Natural Fiber Enhancement of Structural Stability of Composites

The addition of natural fibers to enhance foamed NR not only enables the production of lightweight rubber composites but also imparts optimal strength and structural stability to NR. These improvements were systematically analyzed through a range of material characterization techniques, including DMA, FTIR, XPS, and TGA, performed on pure NR and composites to uncover the underlying modification mechanisms.

The storage modulus reflects the stiffness of a viscoelastic material, indicating its ability to store mechanical energy during loading cycles and serving as a key measure of material resilience. In contrast, the loss modulus quantifies the energy dissipated as heat during loading cycles, providing insight into the damping properties of the material [40].

As shown in Figure 4a,b and Appendix A, the storage and loss moduli of pure NR and composites follow the order CF-NR > BF-NR > DF-NR > AC-NR > Blank. The storage modulus trend indicates that the addition of natural fibers enhances the rigidity and deformation resistance of NR. The increase in loss modulus suggests that a greater proportion of the energy absorbed by the material during deformation is dissipated as thermal energy, thereby improving the comfort of the sole material. These observations indicate that the incorporation of natural fibers restricts the movement of NR molecular chains, thereby enhancing the structural stability of the composite. Additionally, the interfacial friction between the fibers and the matrix facilitates efficient energy dissipation, further improving the durability of the rubber material.

Figure 4c presents the FTIR spectra of pure NR and composites (the assignment of each peak in FTIR is shown in Appendix A). The peaks at 2960 cm^−1^ and 2916 cm^−1^ are attributed to the stretching vibrations of C-H bonds, while the peak at 2850 cm^−1^ corresponds to the symmetric stretching vibrations of −CH_2_ or −CH_3_ groups. The absorption peaks at 1662.73 cm^−1^ and 1583.70 cm^−1^ are assigned to the stretching vibrations of C=O and C=C bonds [41,42], respectively. The broad absorption peak at 3300 cm^−1^ corresponds to the stretching vibration of −OH groups [41,42]. After the addition of AC, the absorption peak at 3300 cm^−1^ increased significantly, indicating a notable increase in hydroxyl content within the material. Other peaks remained nearly unchanged, suggesting that the incorporation of the foaming agent and fibers had minimal impact on the chemical structure of NR. The significant differences in mechanical properties are primarily attributed to the physical support provided by the natural fibers to the foamed pores.

Appendix A and Figure 4e display the XPS C1s, O1s, and S2p spectra of pure NR and composites. The C1s spectra consist of three peaks: the C-C peak at 284.8 eV, the C-O-C peak at 286 eV, and the O-C=O peak at 288.5 eV. The C-S peak is not clearly visible due to the overlap with the C-O-C peak. The O1s spectra show three distinct peaks: the organic C=O peak at 531.5–532 eV, the organic C-O peak at 533 eV, and the H_2_O/−OH peak at 535 eV [43,44,45]. Notably, the proportion of C-O bonds within the rubber increased significantly following the addition of AC, which aligns with the increase in −OH groups observed in Figure 4c.

As presented in Figure 4e, sulfur in all samples exists in two forms: sulfate groups (SO_4_^2−^) at 169 eV and sulfur covalently bonded with organic composites (S-R) at 163.5 eV. For both AC-NR and Blank, the proportion of S-R bonds was around 80%, indicating that not all sulfur atoms participated in the covalent cross-linking network of NR. However, with the addition of natural fibers, the proportion of S-R bonds increased to over 90%. This increase can be attributed to the ability of biomass materials to capture and immobilize sulfur [46,47], thus improving the reaction efficiency of sulfur and facilitating the formation of a denser, more robust cross-linking network in the composites. This enhanced cross-linking density contributes to improved structural stability and strength of the sole material.

Figure 4d and Appendix A present the TG and DTG curves of pure NR and composites (the important values of the TG curves are located in Appendix A). For Blank, there was primarily a single stage of decomposition. However, with the introduction of AC and natural fibers, the thermal decomposition of AC-NR, BF-NR, CF-NR, and DF-NR occurred in three stages. The weight loss between 150 and 300 °C was attributed to the incomplete decomposition of AC components (Appendix A) and the decomposition of active groups on the natural fibers. A second weight loss from 300 to 500 °C was due to the decomposition of NR itself. A further reduction in mass between 600 and 800 °C may be linked to the presence of a small amount of anticaking agent components in the industrialized AC composite foaming agent. The char yield for AC-NR was slightly lower than that of Blank, owing to the near-complete decomposition of AC. Conversely, the char yields for BF-NR, CF-NR, and DF-NR were higher than that of Blank, attributed to both the high char yield of natural fibers (Appendix A) and the increased cross-linking density in the composites containing natural fibers. The temperatures corresponding to the maximum weight loss rate for pure NR and composites were all around 370 °C, suggesting that the primary material, NR, did not undergo significant changes.

Figure 5a displays the fracture surfaces of pure NR and composites. The fracture surface of Blank was flat, indicating that the material distributed the stress uniformly. In contrast, the fracture surface of AC-NR, after the introduction of the foaming agent, exhibited a large number of uniformly dispersed pores. These pores were a result of gas bubbles formed during the decomposition of the AC foaming agent. The surface of AC-NR was uneven, suggesting that tensile deformation caused numerous stress concentration points. The fracture did not occur uniformly throughout the material, which helps explain the reduced strength of AC-NR.

To quantitatively clarify the effects of foaming and fiber addition on materials’ structure, MIP was used to characterize the pore architecture of the composites. The results were displayed in Figure 5b–g. The results indicated that AC-NR exhibits the highest porosity (23.98%) and average pore diameter (60.05 nm), while having the lowest proportion of nano-sized pores (<100 nm), indicating insufficient structural stability of AC-NR. The incorporation of natural fibers led to a simultaneous decrease in both porosity and average pore size within the composites, likely because the fibers physically occupied a portion of the cellular volume. Closer inspection revealed a pronounced rise in the fraction of nanoscale pores (>100 nm), indicating that the pore network originally created by the AC blowing agent was further fragmented by the fibers into a greater number of smaller voids. This not only corroborates the hypothesis that the natural fibers act as heterogeneous nucleation sites but also demonstrates that they interpenetrate the cellular structure, effectively enhancing its stability.

Upon the incorporation of natural fibers, the fracture surface of the composites still exhibited numerous pores, but the surface became relatively smooth. This smoothness indicates that stress was effectively transmitted throughout the material. Furthermore, the natural fibers were interwoven within the pores. This observation supports the hypothesis that natural fibers act as heterogeneous nucleating agents. As a result, the pores tend to aggregate around the fibers, and the fibers provide structural support to the pore network. This mechanism contributes to the enhanced structural stability of the material after the addition of natural fibers. The schematic diagram of this process is presented in Figure 6. Additionally, CF in the CF-NR displayed a highly branched, multi-forked structure (Appendix A), which enabled more robust and efficient support for the pore cavities. This intricate morphology enhances the composite’s structural integrity, making CF-NR perform better than other composites.

### 3.4. Other Properties of Pure NR and Composites

Natural fibers typically exhibit limited thermo-oxidative stability, which may lead to the accelerated aging of the material upon their incorporation. To assess this, all samples underwent exposure to 100 °C for 100 h, followed by an evaluation of changes in tensile strength, tear strength, and elongation at break after thermo-oxidative aging, as shown in Appendix A. Figure 7a,b illustrates the tensile strength retention, tear strength retention, and the changes in elongation at break of pure NR and composites, respectively. Significant reductions in tensile strength, tear strength, and elongation at break were observed across all materials in the accelerated thermo-oxidative aging tests, indicating evident signs of thermo-oxidative aging. The retention rates of tensile and tear strength for Blank were notably low. However, with the addition of AC, both tensile and tear strength retention rates improved, and further enhancement in strength was observed with the inclusion of natural fibers. Thus, the incorporation of natural fibers does not expedite the aging process of NR-based composites.

To elucidate the reasons behind the differences in the thermal-oxidative aging behavior of NR caused by the introduction of natural fibers, the crosslinking degrees and crosslinking densities of pure NR and composites were evaluated both before and after aging. The results were shown in Figure 7c and Appendix A, which indicated that after aging, Blank still maintained a high crosslinking degree, but its crosslinking density decreased. The addition of the AC foaming agent alone led to a reduction in the crosslinking degree and crosslinking density. However, with the addition of natural fibers, this negative impact was mitigated. Moreover, after aging, the crosslinking density of BF-NR, CF-NR, and DF-NR remained higher than that of Blank and AC-NR. Among these three composites, CF-NR still had the highest crosslinking degree and crosslinking density after aging.

Rubber is traditionally favored as a sole material for its exceptional anti-slip properties. However, the inclusion of natural fibers impedes the mobility of NR molecular chains, affecting the material’s anti-slip performance. As shown in Figure 4a, NR-based composites reinforced with natural fibers exhibit an increased loss modulus, particularly at lower temperatures. These notable effects can be attributed to interfacial slippage between the filler and the rubber matrix, as well as internal friction within the rubber phase due to molecular chain movements. To evaluate the impact of natural fiber incorporation on the anti-slip properties of NR, the static friction coefficients of each composite against 304 stainless steel were measured at room temperature, with results presented in Figure 8. Regardless of the type of natural fiber, no significant changes were observed in the static friction coefficient under the testing conditions. Therefore, composites incorporating foaming agents and natural fibers effectively preserve the superior anti-slip properties characteristic of NR.

## 4. Conclusions

Natural fibers have been effectively utilized to improve the structural stability of foamed NR. Compared to pure NR (Blank) and foamed NR (AC-NR), natural fibers effectively addressed the mechanical property degradation typically observed in NR after foaming. Furthermore, the results demonstrated that the incorporation of natural fibers did not compromise the original vulcanization cross-linking network of NR. The enhanced structural stability of NR is primarily attributed to the supporting role of the fibers across the foamed pores. Additionally, the introduction of natural fibers positively influenced the anti-aging properties of NR. The three types of materials selected in this paper each have their own advantages. Bamboo offers significant advantages as a low-carbon material, particularly its rapid growth rate, high fiber strength, and notable cost-effectiveness. Its fast renewability reduces pressure on slower-growing timber resources [48]. CF and DF represent valuable resources recovered from solid waste streams. Utilizing these materials aligns with circular economy principles and stringent low-carbon requirements, as they divert waste from landfills or incineration and reduce the demand for virgin resources [30,49,50]. The economic advantage of these fibers is reinforced by their origin as low-cost or even negative-cost raw materials (waste), which offsets disposal expenses. Herein, this study not only presents an innovative approach to the development of high-quality lightweight footwear in the sole material industry but also efficiently utilizes biomass resources.

## Figures and Tables

**Figure 1 polymers-17-02043-f001:**
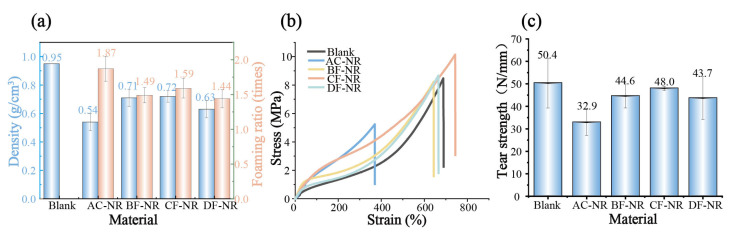
Mechanical properties of pure NR and composites: (**a**) Density and foaming expansion ratio; (**b**) Stress-strain curve; (**c**) Tear strength.

**Figure 2 polymers-17-02043-f002:**
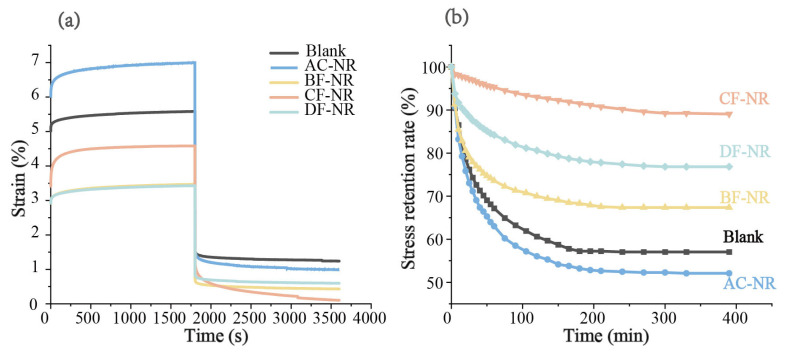
(**a**) Tensile creep-recovery curve of pure NR and composites; (**b**) Stress relaxation curve of pure NR and composites.

**Figure 3 polymers-17-02043-f003:**
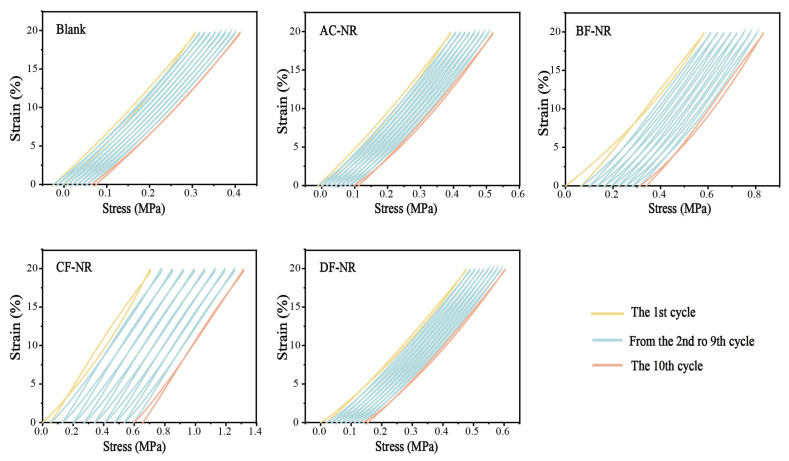
Cyclic stress-strain curve of pure NR and composites.

**Figure 4 polymers-17-02043-f004:**
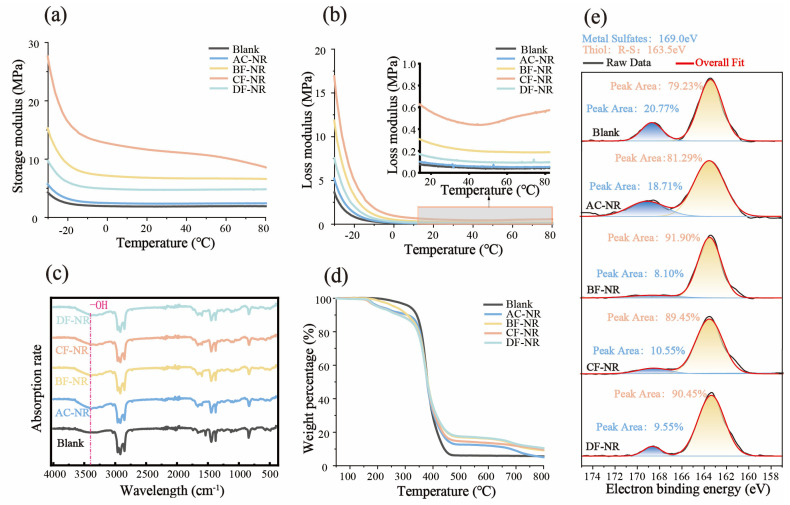
Characterization Results for pure NR and composites: (**a**) Storage modulus; (**b**) Loss modulus; (**c**) FTIR spectra; (**d**) TGA curves; (**e**) XPS S2p.

**Figure 5 polymers-17-02043-f005:**
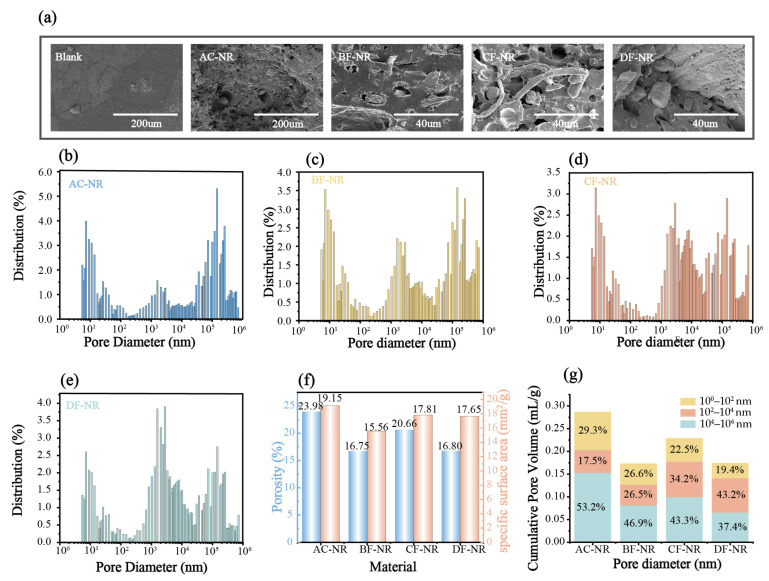
Microstructure of rubber: (**a**) SEM; (**b**–**e**) Pore diameter distributions; (**f**) Porosity and specific surface area; (**g**) Comparison of pore size distribution.

**Figure 6 polymers-17-02043-f006:**
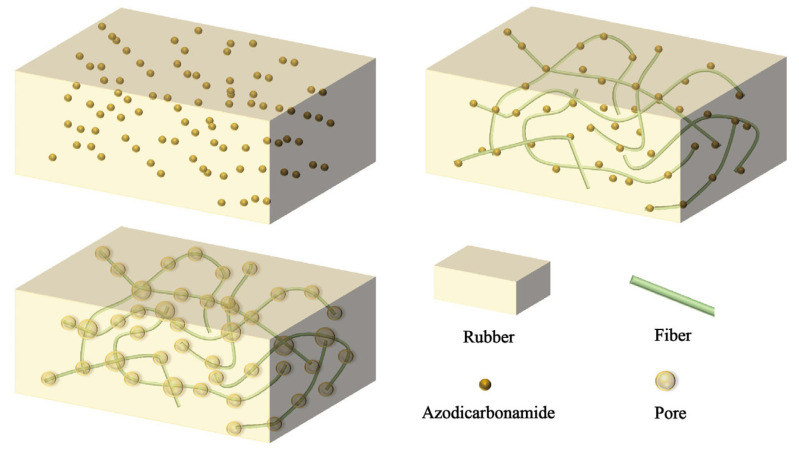
Schematic diagram of foaming mechanism for fiber-reinforced NR-based composites.

**Figure 7 polymers-17-02043-f007:**
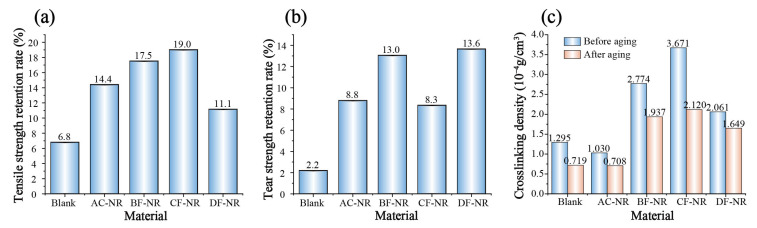
Changes in the strength of pure NR and composites after aging: (**a**) Tensile strength retention rate; (**b**) Tear strength retention rate; (**c**) Crosslinking density.

**Figure 8 polymers-17-02043-f008:**
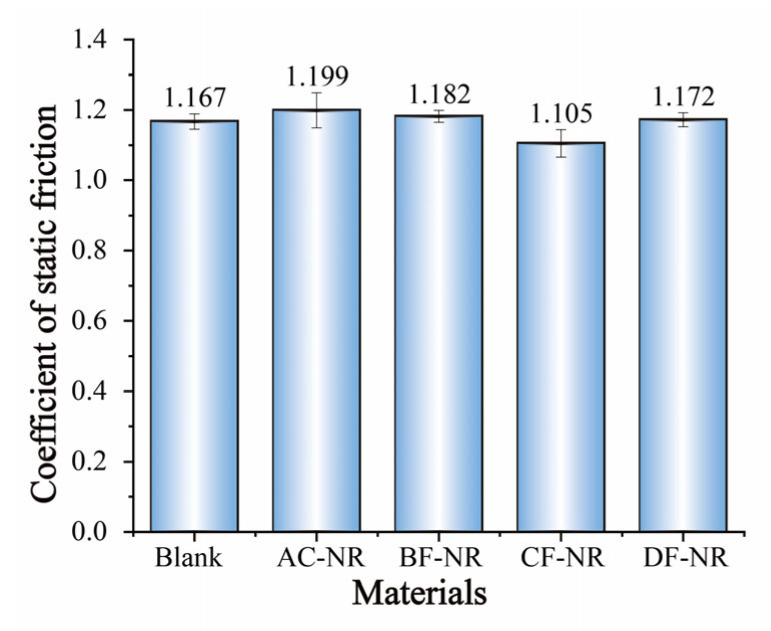
Coefficient of static friction of pure NR and composites.

**Table 1 polymers-17-02043-t001:** Proportions of Raw Materials and Reagents.

Sample	Rubber ^1^	Fiber ^1^	Stearic Acid ^1^	Zinc Oxide ^1^	Sulfur ^1^	Accelerator ^1^	AC ^1^
Blank	100	0	3	5	2	0.2	0
AC-NR	100	0	3	5	2	0.2	20
BF-NR	80	20	3	5	2	0.2	20
CF-NR	80	20	3	5	2	0.2	20
DF-NR	80	20	3	5	2	0.2	20

^1^ The quantity is measured in proportion to weight.

**Table 2 polymers-17-02043-t002:** Deformation values of pure NR and composites.

Sample	${Ɛ}_{KV}^{*}$/%	${Ɛ}_{\infty }^{*}$/%	${Ɛ}_{SM}^{*}$/%	${Ɛ}_{MAX}^{*}$/%	${Ɛ}_{R}^{*}$/%	$\theta_{SK}^{*}$/%
Blank	0.61	1.24	3.73	5.58	4.34	79.39
AC-NR	1.05	0.99	4.95	6.99	6.00	85.84
BF-NR	0.53	0.43	2.51	3.47	3.04	87.46
CF-NR	1.51	0.1	2.97	4.58	4.48	99.33
DF-NR	0.44	0.6	2.39	3.43	2.83	82.51

**Table 3 polymers-17-02043-t003:** Simulated parameters of the Burger’s Model for pure NR and composites.

Sample	*E_M_*/MPa	*E_K_*/MPa	$\eta_{K}$/MPa	$\eta_{M}$/s	*τ*/s
Blank	0.38	8.58	1632.86	24,704.63	190.31
AC-NR	0.32	4.46	620.77	14,998.29	139.19
BF-NR	0.65	7.58	1426.75	25,950.58	188.23
CF-NR	0.52	3.18	322.88	26,157.90	101.53
DF-NR	0.65	8.61	1601.66	27,120.06	186.02

**Table 4 polymers-17-02043-t004:** Simulated parameters of the Weibull distribution function for pure NR and composites.

Sample	$\eta_{r}$/*S*	$\beta_{r}$	${\;Ɛ}_{\mathrm{KV}}$/%	${\;Ɛ}_{\infty }$/%	${\;Ɛ}_{\mathrm{SM}}$/%	${\;Ɛ}_{R}$
Blank	1597.86	0.14	0.85	0.95	3.78	4.63
AC-NR	3575.36	0.10	2.54	0.15	4.30	6.84
BF-NR	20,610.56	0.19	0.81	0.04	2.62	3.43
CF-NR	523.50	0.59	1.08	0.01	3.49	4.57
DF-NR	1592.80	0.27	0.47	0.43	2.53	3.00

## Data Availability

The original contributions presented in this study are included in the article/Appendix A. Further inquiries can be directed to the corresponding author(s).

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
