# Peer review of "Natural Fiber-Reinforced Foamed Rubber Composites: A Sustainable Approach to Achieving Lightweight and Structural Stability in Sole Materials"

_polymers, 2025, doi:10.3390/polym17152043_

Round 1
Reviewer 1 Report
Comments and Suggestions for Authors
Comments to author
- While the study acknowledges that azodicarbonamide (AC) reduces mechanical strength due to pore formation, more mechanistic insights or microstructural evidence (e.g., cross-sectional SEM or porosity quantification) would strengthen this claim.
- The fixed fiber loading of 20 wt% across all composite samples needs stronger justification. Has this been established as optimal in prior studies, or was it chosen arbitrarily?
- Although the SEM images qualitatively support the claim that fibers support the foam structure, a quantitative pore size distribution analysis (e.g., using ImageJ) would substantiate the discussion more rigorously.
- While thermo-oxidative aging is tested at 100°C for 100 hours, comparisons to commercial EVA or PU under similar conditions would contextualise the material's industrial competitiveness.
- Figures such as stress-strain curves and storage modulus plots lack error bars or standard deviation indicators. Repeatability and reproducibility of the results remain unclear.
- Introduction section need to strengthened. Consider these article, “Advancements in natural fibre based polymeric composites: A comprehensive review on mechanical-thermal performance and sustainability aspects’, Quasi-static puncture shear loading characteristics of GLARE/nanoclay laminates with various indenters”, Eco-friendly sugarcane biochar filler for enhanced mechanical properties in S-glass/polyester hybrid composites”.
- The preparation of KH-550 treated fibers is described, but the reproducibility, effectiveness, or uniformity of the silane grafting via FTIR or TGA of fibers alone is not evaluated.
- The manuscript could be improved by including a more explicit comparison of the final composite properties with other bio-based or conventional sole materials, beyond pure NR.
- Some tests were performed at 25 °C and others at 50 °C (e.g., DMA), yet there is no discussion on how temperature variation might affect comparative analysis across tests.
- While Burger’s model and Weibull fitting are conducted, the manuscript does not interpret the physical implications of each parameter (ηM, EK, τ) in depth for material design optimisation.
- XPS data suggest increased S–R bonds due to fiber incorporation, but no direct evidence (e.g., interfacial shear strength or pull-out tests) is provided to evaluate fiber–rubber bonding quality.
- Given the emphasis on sustainability and industrial application, some discussion on the economic feasibility and scalability of using collagen, bamboo, and rice husk fibres is warranted.
- The manuscript notes strength retention post-aging but lacks molecular-level or oxidative degradation explanations.
- Use of collagen and distiller’s grain-derived fibres may pose odour or microbial growth issues in footwear; these potential limitations are not addressed.
- The discussion on static friction coefficient is brief. Would these values remain stable under wet conditions or repeated use? Realistic testing is recommended.
- Although the paper touches on cyclic loading via DMA, long-term fatigue tests (number of cycles to failure) are absent but essential for real-world sole applications.
- The study claims sustainability but provides no life cycle analysis, biodegradability data, or COâ‚‚ savings estimates compared to conventional sole materials.
- The FTIR spectrum interpretation would benefit from a tabulated summary of peak positions, functional group assignments, and corresponding references. While XPS spectra are shown, no atomic concentration percentages or peak area ratios are presented to support conclusions about bonding states or sulfur utilisation.
- The TG analysis focuses mainly on char residue; however, discussing decomposition onset temperature (Tonset) and its relevance to processing or service temperature would be informative.
Reviewer 2 Report
Comments and Suggestions for Authors
Please see attached PDF file

Round 2
Reviewer 1 Report
Comments and Suggestions for Authors
The present form of the manuscript is full consideration for Publication.